# A Review on the Progress, Challenges, and Performances of Tin-Based Perovskite Solar Cells

**DOI:** 10.3390/nano13030585

**Published:** 2023-02-01

**Authors:** Yuen-Ean Lye, Kah-Yoong Chan, Zi-Neng Ng

**Affiliations:** 1School of Electrical Engineering and Artificial Intelligence, Xiamen University Malaysia, Jalan Sunsuria, Bandar Sunsuria, Sepang 43900, Selangor, Malaysia; 2Centre for Advanced Devices and Systems, Faculty of Engineering, Multimedia University, Persiaran Multimedia, Cyberjaya 63100, Selangor, Malaysia

**Keywords:** Sn-based perovskite solar cells, 3G solar cell technologies, lead-free perovskite solar cells

## Abstract

In this twenty-first century, energy shortages have become a global issue as energy demand is growing at an astounding rate while the energy supply from fossil fuels is depleting. Thus, the urge to develop sustainable renewable energy to replace fossil fuels is significant to prevent energy shortages. Solar energy is the most promising, accessible, renewable, clean, and sustainable substitute for fossil fuels. Third-generation (3G) emerging solar cell technologies have been popular in the research field as there are many possibilities to be explored. Among the 3G solar cell technologies, perovskite solar cells (PSCs) are the most rapidly developing technology, making them suitable for generating electricity efficiently with low production costs. However, the toxicity of Pb in organic–inorganic metal halide PSCs has inherent shortcomings, which will lead to environmental contamination and public health problems. Therefore, developing a lead-free perovskite solar cell is necessary to ensure human health and a pollution-free environment. This review paper summarized numerous types of Sn-based perovskites with important achievements in experimental-based studies to date.

## 1. Introduction

The development of various PV technologies throughout the years may be generally categorized into three main generations. First-generation (1G) solar cells are the most common solar cells available on the market, which include single and multi-crystalline silicon. First-generation solar cells have high power conversion efficiencies (PCEs), but their production cost is high too. Second-generation (2G) solar cells based on amorphous silicon, cadmium telluride, and copper indium gallium selenide (CIGS) were developed to further reduce the cost. Nevertheless, the performance of 2G solar cells is often inferior to that of 1G solar cells, thus prompting the development of third-generation (3G) emerging solar cell technologies. Recently, third-generation (3G) emerging solar cell technologies have been popular in the research field as there are many possibilities to be explored, such as dye-sensitized solar cells, copper/zinc/tin sulfide solar cells, quantum dot solar cells, polymer solar cells, organic solar cells, and perovskite solar cells (PSCs) [1,2,3]. PSCs are the fastest-developing technology among these 3G solar cell technologies, which are suitable for generating electricity efficiently with low production costs.

A solar cell that includes a perovskite compound as the light-harvesting active layer is known as a PSC. Usually, a hybrid organic–inorganic lead or tin halide-based material was utilized as the active layer. The typical formula of metal halide perovskites is ABX_3_, where A is a cation, such as organic methylammonium (CH_3_NH_3_^+^ or MA) or formamidinium (HC(CH_2_)_2_^+^ or FA) or alkali metal cesium (Cs^+^) cation; B is a divalent metal ion (e.g., Pb, Sn, and Ge); and X is a halide anion (e.g., Cl, Br, and I) [1]. The A and B cations will coordinate with 6 and 12 X anions, forming octahedral and cuboctahedral geometry, respectively, as shown in Figure 1 [4]. The photoelectric characteristics of metal halide perovskites can be fine-tuned by modifying the A, B, or X site ions or employing multi-component A, B, or X site ions to fulfill specific applications. Perovskite is widely utilized as the light-absorbing material in light-emitting diodes (LED), photodetectors, and solar cells due to its ideal optical characteristics (e.g., light absorption and emission). The perovskite photoactive film is sandwiched between two electrodes, forming a cell architecture [5]. Perovskite materials such as MASnX_3_, FASnX_3_, and CsSnX_3_ are commonly used as sensitizers in Sn-based perovskites. Electron transport materials (ETM) and hole transport materials (HTM) are frequently used between the active layer and the electrodes to facilitate charge transport processes. In general, a transparent conductor oxide (TCO) such as fluorine-doped tin oxide (FTO) is utilized as one of the electrodes. Meanwhile, the top of the cell architecture is deposited with a metal such as silver or gold. Mainly, there are two distinct charge collection strategies from PSC devices: conventional (n-i-p) and inverted (p-i-n) structures (see Figure 2) [5]. The main difference between these two architectures is that the current flows in the opposite direction [5].

Next, the power conversion efficiency (PCE) of PSCs has been improved from an initial 3.8% to a certified value of 29.8% in 2021, which fulfills the requirement for commercialization [6]. The PSCs’ excellent efficiency, ease of fabrication, and the possibility to substitute on flexible substrates make them particularly intriguing for photovoltaic (PV) energy conversion. In addition, perovskites have unique characteristics that make them suitable for PV applications and preferable to other workaround techniques, such as low defect densities, sharp optical absorptions, solution processability, excellent charge carrier properties, and bandgap tunability [6,7,8,9,10].

However, several issues regarding technological and material challenges remain to be addressed. For example, the short lifespan of PSCs, large-scale manufacturing, and lead toxicity inhibit perovskite PV technology’s commercialization and mass deployment. Several studies have focused on enhancing the stability of the perovskites with potential solutions. For instance, research from Marshall et al. has improved the stability and efficiency of PSCs without a hole-selective interfacial layer [11]. Next, by alloying Ge (II) in CsSnI_3_ to develop a CsSn_0.5_Ge_0.5I3_ composition perovskite, thin films of CsSnI_3_-based PSCs can become very stable and air tolerant [12]. These PSCs have less than 10% decay in efficiency after 500 h of continuous operation in an N2 environment under one-sun illumination. Qiu et al. also suggested creating high-quality B-γ CsSnI_3_ thin films with a two-step sequential deposition process [13]. Furthermore, to overcome the difficulty of large-scale manufacturing, several researchers have focused on refining deposition techniques and enhancing perovskite nanocrystals or perovskite ink [14,15,16].

There has been much research indicating the lead in PSC was able to be replaced by other metals with similar semiconducting properties, such as tin (Sn), bismuth (Bi), germanium (Ge), and antimony (Sb) [17,18,19,20]. Sn (II)-based halide perovskites have exhibited a significant PCE among these alternatives, attracting the most interest in the PSC field. As such, a detailed review of Sn-based perovskites will be discussed in the following section.

The common Sn-based perovskites that can be seen are methylammonium tin iodide (MASnI_3_), formamidinium tin iodide (FASnI_3_), and cesium tin iodide (CsSnI_3_). These Sn-based perovskites have a narrower and more appealing direct bandgap than Pb-based PSCs, which are 1.20 eV, 1.41 eV, and 1.3 eV for MASnI_3_, FASnI_3_, and CsSnI_3_, respectively [21]. Under ambient conditions, the Sn^2+^ will deteriorate to Sn^4+^ and oxidize to an environmentally friendly material, SnO_2_, making it suitable to use as a substitute for lead. Sn-based perovskites and Pb-based perovskites reveal a striking similarity when comparing the major physical parameters [22].

Many papers have focused on Sn-based PSCs since the studies carried out by Noel et al. and Hao et al. obtained a PCE of about 6% [23,24]. The recent progress of Sn-based PSCs has been observed to deliver relatively high performance (PCE > 14%) [25,26,27]. However, due to their narrow bandgaps, Sn-based PSCs often exhibit high short-circuit current densities (J_SC_) of 20 to 25 mA cm^−2^. Another point to notice is that the average open-circuit voltage (V_OC_) of the solar cells is only approximately 0.5 V, far lower than the 1.1 V of the Pb-based perovskites. The tendentious and undesirable oxidation of Sn^2+^ to Sn^4+^ causes the materials to have heavy p-type doping, which behaves as a p-type dopant in the structure and leads to very high photocarrier recombination and excessively high dark-carrier concentration. There has been a lot of effort contributed to enhanced photovoltaic performance and stability of Sn-based PSCs [28,29,30]. The following subsections will discuss the notable achievements in Sn-based perovskites’ experimental studies.

## 2. Methylammonium Tin Halides (MASnX_3_)

### 2.1. Characteristics of MASnI_3_

The literature on the optoelectronic characteristics of MASnI_3_ with a unique method, for introducing spin–orbit coupling (SOC) effects into their efficient GW system, was highlighted by Umari et al. [31]. The details of the comparison is tabulated in Table 1. It can be observed that MASnI_3_ has higher charge mobility, a smaller direct bandgap at the Γ point, and an absorption coefficient in the visible region higher than MAPbI_3_. Researchers were prompted to pay closer attention to Sn-based perovskites because of these pleasing results.

### 2.2. Enhancement in MASnI_3_

Unfortunately, the poor homogeneity and low coverage of Sn-based perovskites are critical problems to overcome. This has led to low PCEs being generated in direct contact with the ETL and the HTL. The film-forming techniques suited for Pb-based perovskites have become ineffective for Sn-based perovskites because the Sn-based perovskites have a quicker crystallization rate [32,33]. Therefore, various research has established the impact of solvents on the crystallization of MASnI_3_ perovskite films [32]. The homogeneous, pinhole-free perovskite film can be prepared with a transitional SnI_2_·3DMSO intermediate phase from a dimethyl sulfoxide (DMSO) solution. A schematic illustration of the formation of CH_3_NH_3_SnI_3_ perovskite film starting from SnI_2_ through the SnI_2_·3DMSO intermediate is shown in Figure 3. This elevated perovskite film creates heterojunction depletion solar cells with a photocurrent of up to 21 mA cm^−2^ without any HTL. Transient photovoltage decay and charge extraction experiments indicated that the MASnI_3_ perovskite device has higher carrier densities and similar recombination lifetimes than MAPbI_3_-based devices [32].

Many methods have been proposed to produce high-quality Sn-based perovskite films [34,35]. For instance, the physical vapor deposition layers of tin (II) iodide (SnI_2_) were transformed to MASnI_3_ by reacting with a spin-coated methylammonium iodide (MAI) solution. This approach produced perovskite particles larger than 200 nm with a complete surface covering. This film was exceptionally stable, making it significant as an absorber layer in the device [35]. Another point to notice is that the morphologies of the films were strongly reliant on MAI concentrations. According to the research from Weiss et al., larger and more homogeneous MASnI_3_ crystals were produced as the MAI concentration increased [35]. A high-quality, dense film with considerably increased stability was obtained after 10-min thermal annealing. The films remained stable under both light and dark conditions after 90 min of air exposure. 

### 2.3. Performance of MASnX_3_

The performance of MASnX_3_ is tabulated in Table 2. The performance of MASnI_3_ can be improved with the addition of Br anion, forming the MASnI_3-x_Br_x_ structure. As a result, there is an increase in V_OC_ from 0.68 V to 0.88 V when transitioning from pure iodide to pure bromide perovskite. This significant improvement in V_OC_ also led to an increase in FF, from 48% to 59%. Nevertheless, there was a drop in J_SC_, where it fell from 16.30 mA cm^−2^ for MASnI_3_ to 8.26 mA cm^−2^ for MASnBr_3_. Among the MASnI_3-x_Br_x_ structure, MASnIBr_2_ has the best performance, where it has a V_OC_ of 0.82 V, J_SC_ 12.3 mA cm^−2^, FF of 57%, and PCE of 5.73%. The device performance from Hao, Stoumpos, Chang et al. was slightly better than Hao, Stoumpos, Cao et al., although they have the same device structure [24,36]. This is because the preparation for device fabrication is different; the thickness of materials and additives are the factors affecting device performance. Lastly, the PCE of MASnI_3_ increased to 6.83% in 2019 when different device structures were applied. Thus, optimization in the device structure design and interface engineering will improve the overall performance of MASnX_3_.

## 3. Formamidinium Tin Halides (FASnX_3_)

### 3.1. Characteristics of FASn(I,Br)_3_

FASnI_3_ is a three-dimensional perovskite having a direct bandgap of 1.41 eV. Another point worth mentioning is that various halides can be used to modify the bandgaps of FASnX_3_. For example, FASnI_2_Br has a direct bandgap of 1.68 eV, and FASnBr_3_ has a direct bandgap of 2.4 eV. The threshold charge-carrier density of the FASnI_3_ material is 8 × 10^17^ cm^−3^, and the charge-carrier mobility is 22 cm^2^V^−1^s^−1^. Next, FASnI_3_ has a lower conductivity than MASnI_3_, but they have similar thermal stability [38]. Mitzi and Liang carried out the first research on the synthesis and characterization of cubic FASnI_3_ perovskite in 1997. Due to their better air stability, FASnI_3_ materials have been used in most high-performance Sn-based PSCs recently [28,39,40]. An experiment conducted by Wang et al. has proved that the FASnI_3_ trap density is as low as 10^11^ cm^−3^, which can prevent water and oxygen from entering the crystallization of FASnI_3_ crystal [41]. As a result, the FASnI_3_ crystal has better air stability than the MASnI_3_ crystal [41].

### 3.2. Enhancement in FASnI_3_

A previous study has verified that the SnF_2_ additive had a similar effect on the FASnI_3_-based solar cells’ performance [21]. The addition of SnF_2_ has a comparable impact on inhibiting Sn^2+^ oxidation to Sn^4+^ and decreasing background carrier density. The FASnI_3_ PSC with 20% of SnF_2_ is able to yield a PCE of 2.1% [42]. Nevertheless, a higher concentration of SnF_2_ will cause significant phase separation in FASnI_3_ perovskites film, resulting in solar cell performance deterioration. Lee et al. have introduced a workaround, that is, adding pyrazine to a precursor solution, including DMF/DMSO, to produce high-quality FASnI_3_ films with good coverage and a uniform surface [43]. Pyrazine doping can greatly minimize phase separation and eliminate Sn vacancies effectively because the N atoms in pyrazine are able to accept lone pairs of electrons, allowing for the formation of a uniform, dense, and pinhole-free FASnI_3_ perovskite layer with high reproducibility and a good PCE of 4.8% [44].

Additionally, the encapsulated solar cells displayed good lasting stability, preserving 98% of their initial PCE after being stored under ambient conditions for more than ten days. Besides the addition of SnF_2_ and pyrazine, various antioxidants were suggested to be applied in Sn-based perovskites in further research; for instance, by adding hydrazine iodide, hydrazine vapor, and Sn powder [37,45]. These antioxidants operate similarly to SnF_2_, increasing the stability and effectiveness of Sn-based PSCs by lowering Sn vacancies and background carriers in Sn perovskite. 

On the other hand, other unique compounds have been employed as additions to improve the perovskites’ film quality and keep Sn^2+^ from oxidizing. For example, it has been suggested to employ the bidentate ligand 8-hydroxyquinoline (8HQ) as a perovskite additive. The 8HQ’s N and O atoms may pair with Sn^2+^ simultaneously, inhibiting Sn^2+^ from oxidizing [46]. Meanwhile, the synthesis of the complex increased the quality of the FASnI_3_ film and decreased non-radiative recombination caused by defect states. It has been demonstrated that adding ammonium hypophosphite to the FASnI_3_ perovskite precursor inhibits Sn^2+^ oxidation and aids the formation of perovskite particles. As a result, the overall quality of the perovskite film has improved with a lesser defect density, and a PCE of 7.3% has been achieved for the final device [17]. It is worth mentioning that roughly half of their initial PCE was retained after 500 h exposed to the air. Next, Wu et al. have adopted a π-conjugated Lewis base molecule that has high electron density to precisely tune the FASnI_3_ perovskite’s crystallization rate [47]. A dense and homogeneous perovskite film with a significantly improved carrier lifetime was achieved by establishing a secure intermediate phase in the Sn-I frameworks. Moreover, the addition of the π-conjugated system inhibited moisture permeability into perovskite crystals, causing the film deterioration to reduce substantially. The perovskite solar cell can maintain over 90% of its initial value with a PCE of 10.1% after 1000 h of light exposure in the air.

Next, the addition of ethylenediammonium (en) cations into Sn-based perovskites led to new 3D hollow perovskites with a tuned direct bandgap from 1.3 eV to 1.9 eV. As such, the V_OC_ and PCE of this perovskite with 10% en addition were much more significant than those of normal ASnI_3_ perovskites [29]. Adding en may increase carrier lifetime, improve film morphology, and decrease background carrier density, contributing to the improvement of Sn-based solar cells’ PCE and stability. Adding en into the FASnI_3_ device can obtain a 7.14% of PCE with greater V_OC_ and FF based on this viewpoint. Similar findings were found in MASnI_3_ and CsSnI_3_ solar cells. A study from Ke et al. has addressed the formation of novel hollow perovskites of TNFASnI_3_ and PNFASnI_3_ by two additional cations, which are diammonium cations of trimethylenediammonium (TN) and propylenediammonium (PN) [48]. The device performance was similarly enhanced by TN and PN, which have larger sizes than en. PCEs of 5.85% and 5.53% were attained utilizing FASnI_3_ absorbers combined with 10% PN and 10% TN, respectively [48].

### 3.3. Performance of FASnX_3_

The performance of FASnX_3_-based PSCs is tabulated in Table 3. The experimental studies have shown that the addition of Br components and additives (SnF_2_) can decrease the hysteresis of the film and improve its stability. However, these methods have obtained very low PCEs which are 1.72% and 2.1%, respectively. Another point to notice is a higher concentration of SnF_2_ will cause significant phase separation in FASnI_3_ perovskites film, resulting in the PCE of FASnX_3_ decreasing from 5.59% to 3.35% when the concentration of SnF_2_ is increasing [39]. Furthermore, there was a significant improvement in the performance of FASnI_3_ when TMA, en cations, or AHP antioxidants were added, where they obtained a similar PCE range from 7.09% to 7.34%. Lately, the introduction of π-conjugated Lewis base molecules and LFA solvent to the FASnI_3_ has further improved the development of perovskite equilibrium structures and stabilities with an efficiency of over 10%. To summarize, the introduction of various solvents, antioxidants, and deposition methods was the main factor affecting the performance of the FASnI_3_ device. 

## 4. Cesium Tin Halides (CsSnX_3_)

### 4.1. Characteristics and Enhancement in CsSn(I,Br)_3_

Inorganic CsSnX_3_ compounds were first synthesized and analyzed by Scaife et al. [51]. However, using CsSnX_3_ compounds was not that popular until Chen et al. revealed the use of CsSnI_3_ in PSCs in 2012 [52]. The CsSnI_3_ black film was synthesized under thermal annealing and fabricated by depositing CsI and SnCl_2_ alternately on a glass substrate. The bandgap and PCE of the CsSnI_3_ PSCs obtained were 1.3 eV and 0.9%, respectively. Further on, Kumar et al. proposed a device that has high photocurrent output and lower Sn vacancies, that is, CsSnI_3_ PSC combined with SnF_2_ [53]. With 20 mol % of SnF_2_, the J_SC_ of 22.70 mA cm^−2^, FF of 0.37, and V_OC_ of 0.24 V were obtained, resulting in an optimal PCE of 2.02%. Furthermore, the devices had less hysteresis than their non-SnF_2_ counterparts. Meanwhile, the measurements of incident photon-to-current conversion efficiency (IPCE) designated that the onset wavelength increased to 950 nm, corresponding to a band gap of 1.3 eV.

The following year, Sabba et al. indicated the CsSnX_3_ compound was a semiconductor with a high propensity to self-doping due to the hole carriers generated from the oxidation of Sn^2+^ to Sn^4+^ [54]. CsSnI_3_ has a similar optical absorption coefficient to MAPbI_3_ (10^4^ cm^−1^) with a low exciton binding energy of 18 × 10^−3^ eV. Moreover, it was a three-dimensional p-type orthorhombic perovskite structure with a bandgap of 1.3 eV [55,56,57]. As a result, it could be used as a light-absorbing material for Pb-free PSCs. The most notable limitation to developing a CsSnI_3_ perovskite solar cell is its instability. The black phase CsSnI_3_ may easily be transformed to the yellow phase CsSnI_3_ in the ambient due to oxidation [44]. As mentioned before, the addition of excessive SnI_2_ to CsSnI_3_-based PSCs is able to increase their efficiency and stability [11].

Broad surface coverage and low defect density CsSnI_3_ films were developed at high temperatures or in an Sn-rich environment. In the presence of 10 mol % excessive SnI_2_, the configuration device’s PCE of ITO/CuI/CsSnI_3_/fullerene/bathocuproine (BCP)/Al grew from 0.75% to 1.5%, and FF and V_OC_ remained satisfactory after 20 min in an illuminated environment. The J_SC_ decline rate was significantly reduced even though the J_SC_ immediately degraded by 10% in the first 50 min. Due to a suitable vacuum level shift, the excess SnI_2_ at the CsSnI_3_/CuI interface created an interfacial dipole that operated as an HTL from CsSnI_3_ to CuI [11]. 

Next, several additives were added to the precursor to enhance the film quality of the CsSn(I, Br)_3_ perovskite. The performance of CsSn(I, Br)_3_ PSCs has improved dramatically with various Sn halide additions, including SnF_2_, SnCl_2_, SnI_2_, and SnBr_2_ [58,59]. Marshall et al. have found that 10 mol % SnCl_2_ in the HTL-free devices had the most excellent FF and PCE as it has the greatest pinhole density [60]. Developing an extremely thin hole-selective layer of SnCl_2_ at the ITO/CsSnI_3_ junction mainly contributed to these results. Doping Br into CsSnI_3_ was also recommended, as the CsSnI_3_-xBr_x_ has a negligible overlayer [54]. The Br-doped CsSnI_3-x_Br_x_ perovskites displayed a substantially higher FF than CsSnI_3_. As the Br component increases, the crystal structure will transform from orthorhombic (CsSnI_3_) to cubic (CsSnBr_3_). The optical bandgap edge start was also changed from 1.27 eV (CsSnI_3_) to 1.37 eV, 1.65 eV, and 1.75 eV, respectively, for CsSnI_2_Br, CsSnIBr_2_, and CsSnBr_3_. Because of their superior thermal and air stabilities, CsSnI_2_Br, CsSnIBr_2_, and CsSnBr_3_ were acceptable for solar cell applications [54].

Furthermore, Qiu et al. also suggested creating high-quality B-γ CsSnI_3_ thin films with a two-step sequential deposition process [13]. The proposed device obtained a bandgap of 1.48 eV and a high absorption coefficient. By improving material quality and device engineering, inorganic perovskite solar cells with higher efficiency and superior stability may be predicted based on the bandgap and the Shockley–Queisser limit [13]. In 2021, a phthalimide (PTM) additive was introduced to reduce the relatively grain-ordered perovskite film and defect density [61]. The performance of B-γ CsSnI_3_-based PSCs has increased to 10.1% and maintained at 94.3% under inert (60 days), 83.4% under ambient (45 days), and 81.3% under one Sun continuous illumination at 70 °C conditions [61].

### 4.2. Performance of CsSnX_3_

The performance of CsSnX_3_ is tabulated in Table 4. One of the highlights is when 20 mol % of SnF_2_ was added to the CsSnI_3_, a high photocurrent output device was generated, where J_SC_ of 22.70 mA cm^−2^ is obtained, followed by V_OC_ of 0.24 V, FF of 37%, and PCE of 2.02%. Moreover, the strategy of adding SnI_2_ to the CsSnX_3_ precursor has simultaneously enhanced the PCE (2.1%) and stability of the device in air exposure. In the following year, the performance of CsSnI_3_ increased drastically as the simple solvothermal process was first introduced in the device fabrication. The PCE of CsSnI_3_ has increased to 12.96%, with a favorable V_OC_ of 0.86 V, J_SC_ of 23.2 mA cm^−2^, and FF of 65%. The PCE of B-γ CsSnI_3_-based PSCs from Qiu et al. in 2017 was lower than 1%, but its pronounced stability and stable conversion efficiency have prompted further exploration of possibilities to obtain high-quality perovskite solar cell films [13]. Therefore, the performance of B-γ CsSnI_3_-based PSCs with PTM additive has greatly improved with a V_OC_ of 0.64 V, J_SC_ of 21.81 mA cm^−2^, and FF of 72.1%, and PCE of 10.1%.

## 5. Low-Dimensional Sn-Based Perovskite

### 5.1. Characteristics of Low-Dimensional Sn-Based Perovskite

There are several large cross-sectional studies that suggest large spacer cations to further improve the stability of Sn-based perovskites. As these organic cations were large, they did not fit into the cubic structure’s vacancies, resulting in the formation of new materials and modifications to the initial perovskite characteristics, which are low (quasi-2D) and mixed (2D/3D) dimensional perovskite structures. A schematic illustration of 2D perovskite and 2D/3D mixture perovskite is shown in Figure 4 and Figure 5, respectively. Two-dimensional halide perovskites were discovered as a solution for stabilizing some metastable phases and as a significant relief from the limited compositional variety. The 2D perovskite (A′)_m_(A)_n-1_B_n_X_3n+1_ crystal lattice adopts a new structural and compositional dimension, A′, where monovalent (m = 2) or divalent (m = 1) cations can intercalate between the anions of the 2D perovskite sheets [63].

### 5.2. Enhancement in Low-Dimensional Sn-Based Perovskite

Lately, the existence of large organic ammonium cations such as phenylethylammonium (PEA), butylammonium (BA), 2-hydroxyethylammonium (HEA), guanidinium (GA), and more has been demonstrated to resist moisture intrusion at the Pb-based perovskite nanolayers’ boundaries, making the films and devices have unprecedented high stability. The major goal of exploring quasi-2D and 2D/3D mixed perovskites for PV systems has been to enhance their stability against moisture owing to the hydrophobic nature of the bulky cations [65,66]. The insertion of varied quantities of organic amine molecules allows the optical bandgap of perovskite to be adjusted to ideal energy shells, allowing perovskite materials to be applied in more applications. 

Due to the low chemical stability of tin (Sn^2+^) in the perovskite lattice, it is difficult to regulate the film morphology as well as the Sn vacancies in the perovskites. As a consequence, the performance of Sn-based PSCs has increased significantly slower than that of Pb-based PSCs. In 2019, Wang, Dong, et al. showed that enhancement of device stability and reduction of Sn-vacancies and other defect densities can be achieved by utilizing quasi-2D Sn-based perovskite as an absorber layer in the device [67]. Furthermore, Gai et al. have introduced a method that involves injecting a tiny number of big organic cations into a 3D precursor in the development of 2D/3D mixed Sn-based perovskites to obtain a far better performance and stability than pure quasi-2D [68]. The advantages of 2D/3D mixed heterostructures were that they preserved the features of 2D structures while exhibiting the characteristics of 3D structures. Other benefits of quasi-2D and 2D/3D mixed perovskites include a high-quality film created by one-step spin-coating without high-temperature annealing, a big bang gap resulting in a high open-circuit voltage, and more adjustable architectures than their 3D counterparts [69]. In addition, quasi-2D hybrid perovskites do not require following the Goldschmidt factor for the organic cation R-NH_3_^+^, so they will have higher chemical structural flexibility than 3D perovskites [69]. Because of quasi-2D hybrid perovskites’ unique features such as higher formation energy and hydrophobicity, they are considerably more moisture resistant, so the devices may be produced in a humid environment.

Unfortunately, the appearance of isolating long-chain organic cations also causes anisotropy in the crystal, which will affect device performance significantly. As a result, the greater stability of the quasi-2D PSCs comes at the cost of decreased performance since these insulating organic spacers hinder vertical charge transport [70,71]. Under this circumstance, these inorganic sheets will align perpendicular to the substrate, so a highly vertically oriented perovskite membrane is considered a prerequisite for overcoming the problem of quasi-2D PSCs’ inefficiency [72]. Several techniques for generating very vertically oriented membranes have been established in Pb-based quasi-2D perovskites. Still, due to differences in characteristics between Pb- and Sn-based perovskites, most are inappropriate for Sn-based perovskites [71]. The research from Cao et al. has shown that the thin films of 2D perovskites orient the {(CH_3_NH_3_)_n-1_Sn_n_I_3n+1_}^2-^ slabs can be flipped perpendicular to the substrate with N, N-dimethylformamide as a solvent for deposition [70]. Next, to regulate the crystallization process, Qiu et al. combined n-butylamine (BA) and PEA organic cations into a 2D Ruddlesden–Popper (2D RP) Sn perovskite, generating extremely vertically aligned [(BA_0.5_PEA_0.5_)_2_FA_3_Sn_4_I_13_] 2D RP perovskites [73]. The PCE of [(BA_0.5_PEA_0.5_)_2_FA_3_Sn_4_I_13_] 2D RP perovskites were enhanced to 8.82% because of the combination. Furthermore, Liao et al. have altered the perovskites’ orientation domains by adjusting the ratio of PEA/FA [74]. The addition of 20% PEA resulted in a greatly oriented perovskite membrane perpendicular to the substrate. Therefore, adding 20% PEA to FASnI_3_ perovskite solar cells will attain a maximum PCE of 5.94% while preserving improved stability. 

Since quasi-2D perovskites have exceptional physical and chemical characteristics, they have been discovered to play an essential role in enhancing the stability of PSCs under ambient environments.

### 5.3. Performance of Low-Dimensional Sn-Based Perovskite 

The performance of low-dimensional Sn-based perovskite is tabulated in Table 5. In 2019, it can be noticed that mixed spacer organic cations (BA and PEA) in 2D RP Sn PSC have achieved a promising PCE of 8.82% with improved crystal orientation and high-quality film morphology. Due to their properties of low toxicity and superior stabilities, 2D RP Sn PSCs are crucial to the commercialization of perovskite-based PV systems. Next, the performance of Sn-based perovskite has been further improved by enhancing its electron-transporting layer design. The utilization of a higher energy level indene-C_60_ bisadduct in the device architecture has achieved an incredibly high V_OC_ of 0.94 V, resulting in a considerably high PCE of 12.4%.

## 6. Mixed A Cations Sn-Based Perovskites

### 6.1. Characteristics of Mixed A Cations Sn-Based Perovskites

Metal halide perovskites with mixed cations were broadly used in Pb-based PSCs. A schematic illustration of mixed A cations Sn-Based perovskites is shown in Figure 6. These PSCs exhibit high efficiency and good stability because of the mixture-triggered film morphology improvement, suppressing carrier recombination within the device, and enhancing water and oxygen resistance [76,77]. In this circumstance, developing stable and highly efficient Sn-based PSCs with mixed-cation perovskites is also a good strategy. Previous research has reported an improved efficiency of up to 10.07% by mixing MA and Cs cations into Sn-based perovskites with MA_0.9_Cs_0.1_SnI_3_ PSCs [78]. In 2017, Zhao et al. developed 3D Sn-based perovskites with mixed MA and FA cations [59]. It was found that optimizing the ratio of mixed cations altered the optical property significantly. Thus, adding 10 mol % of SnF_2_ into (FA)_0.75_(MA)_0.25_SnI_3_ PSC can achieve a maximum PCE of 8.12%.

### 6.2. Performance of Mixed A Cations Sn-Based Perovskites 

The ratio of mixed A cations plays an important role in altering the optical property of PSC. It can be observed from Table 6 that the (FA)_0.75_(MA)_0.25_SnI_3_ with 10% of SnF_2_ has the best PCE of 8.12%, followed by V_OC_ of 0.61 V, J_SC_ of 21.2 mA cm^−2^, and FF of 62.7%. This composition engineering is one of the important strategies for increasing the Voc and PCE of Sn-based PSCs. 

## 7. Current Challenges of Sn-Based Perovskites

### 7.1. Commercialization of PSCs

The commercialization of solar cell devices is determined by three factors: (a) device performance, (b) stability, and (c) cost. When a device is in use, its operational stability affects its lifespan, emphasizing a variety of operational pressures and environmental stresses that accelerate device aging, such as mechanical load, heat, humidity, and more. Operational stresses are inextricably linked to the working device, unlike environmental issues, which may be effectively avoided by adding external protection such as encapsulation [79]. Thus, it is crucial to strengthen the intrinsic robustness of PSC to obtain adequate operational stability. 

The requirements for large-scale commercialization are that PCE and stability must meet the industry standard through the test conducted by the International Electrochemical Commission (IEC) following IEC 61215 regulations [80]. There are three main categories to test the stability: thermal cycling test, humidity-freeze test, and damp heat test [80]. These methods could validate the long-term stability and device performance of the developed PV module and are important to win the confidence of investors and consumers, which leads to PV module commercialization.

### 7.2. Fabrication Challenges of Sn-Based PSCs

Tin was discovered to be a viable substitute for lead in the PSC. However, this approach has not yielded a comparable performance of Sn-based lead-free PSCs to the lead counterparts. Modifications in device architecture, improvements to the absorber layer, inclusion of new carrier transport materials (ETLs and HTLs), and more have become a path to enhance the efficiency of Sn-based PSCs recently. In the fabrication of the film, precise control over the grain structure, stoichiometry, and crystallographic phase of the perovskite layer is necessary to develop an efficient PSC. This is because these characteristics are highly dependent on the film deposition process. Future work on developing different fabrication procedures is suggested to produce efficient PSCs for large-scale implementation.

### 7.3. Stability Enhancement of Sn-Based PSCs

The main challenges affecting the performance of the Sn-based PSCs are the oxidation of Sn^2+^ and external stresses such as illumination, moisture, oxygen, and thermal instability. To obtain high purity of the Sn source, the presence of Sn^4+^ must be eliminated during the fabrication of Sn-based PSCs. Many efforts have been made to resolve this issue, such as adding additives to enhance the overall device performance. The solvent SnI_2_ was one of the additives commonly used in Sn-based alloys to reduce the impurity of the Sn source. Next, the fast crystallization rate of Sn-based perovskites has led to poor homogeneity of the films. Hence, controlling the crystallization and nucleation process is important to reduce rough and defective films. To produce a high-quality film, the perovskite crystallization and nucleation processes can be regulated with the introduction of the intermediate phase (e.g., DMSO and π-conjugated Lewis base), templated growth, Ostwald ripening effect, and more [32,47,81,82].

Moreover, the notable achievements in low bulk defect and high efficiency have gained a lot of attention for future research. This is because the characteristics and stability of Sn-based PSCs were modified with the use of large spacer cations. Large organic ammonium cations such as phenylethylammonium (PEA) and butylammonium (BA) are employed to form 2D perovskites. However, the charge transport in the 2D perovskite caused the efficiency to decrease, so a 3D precursor was introduced to develop 2D/3D mixed Sn-based perovskites to obtain far better performance and stability than pure quasi-2D perovskites [69]. Lastly, mixed A cations were another approach to improving overall device performance, as the mixture can improve the film morphology, suppress carrier recombination within the device, and enhance water and oxygen resistance [76,77]. These promising strategies have indicated that the stability and performance of Sn-based perovskites can be improved in the future.

## 8. Conclusions

Energy shortages have prompted the development of environmentally friendly and cost-effective substitutes to the usage of fossil fuels. Among the 3G solar cell technologies, the PSCs are the fastest-developing technology, which is suitable for generating electricity efficiently with low production costs. The PCE of PSCs has witnessed a tremendous increase in efficiency in a short period, rising from an initial 3.8% to a certified value of 29.8% in 2021. Although Pb-based perovskite solar cells have outstanding optoelectronic properties, mass production of PSCs has been hindered due to their toxicity and poor stability. Therefore, researchers are working on substituting lead with less toxic compounds while improving the stability and maintaining the efficiency of the PSCs. Various Pb-free PSCs have made significant progress, including tin (Sn), bismuth (Bi), germanium (Ge), and antimony (Sb). Among these alternatives, tin (Sn) is the most promising as it has excellent optoelectronic properties such as narrow bandgaps, high photocarrier recombination, and an excessively high dark-carrier concentration, making it suitable to substitute the Pb-based PSCs. Despite Sn-based PSCs’ remarkable PV performance over the last decade, the problem of maintaining their stability remains a challenge that needs to be resolved. Additives, intermediate phase, partial substitution, and chemical engineering are some of the approaches and opportunities to enhance the performance of PSC devices. It would be immensely beneficial to conduct more research on the dimensionality in enhancing Pb-free PSC efficiency and stability. Additionally, more research on the characteristics, defect properties, and charge transport of Pb-free perovskites is required to make commercialization possible.

## Figures and Tables

**Figure 1 nanomaterials-13-00585-f001:**
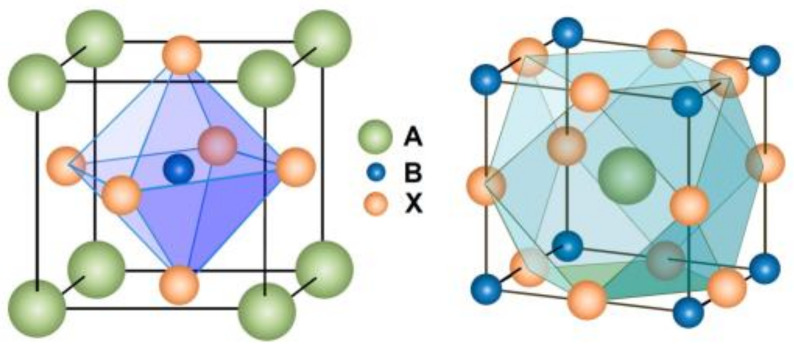
ABX_3_ perovskite structure showing (**left**) BX_6_ octahedral and (**right**) AX_12_ cuboctahedral geometry. Reproduced with permission from [4]. Copyright American Chemical Society, 2014.

**Figure 2 nanomaterials-13-00585-f002:**
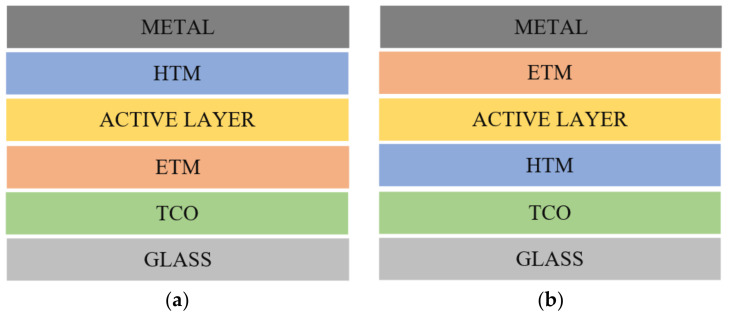
(**a**) Conventional (n-i-p); (**b**) inverted (p-i-n) structure.

**Figure 3 nanomaterials-13-00585-f003:**
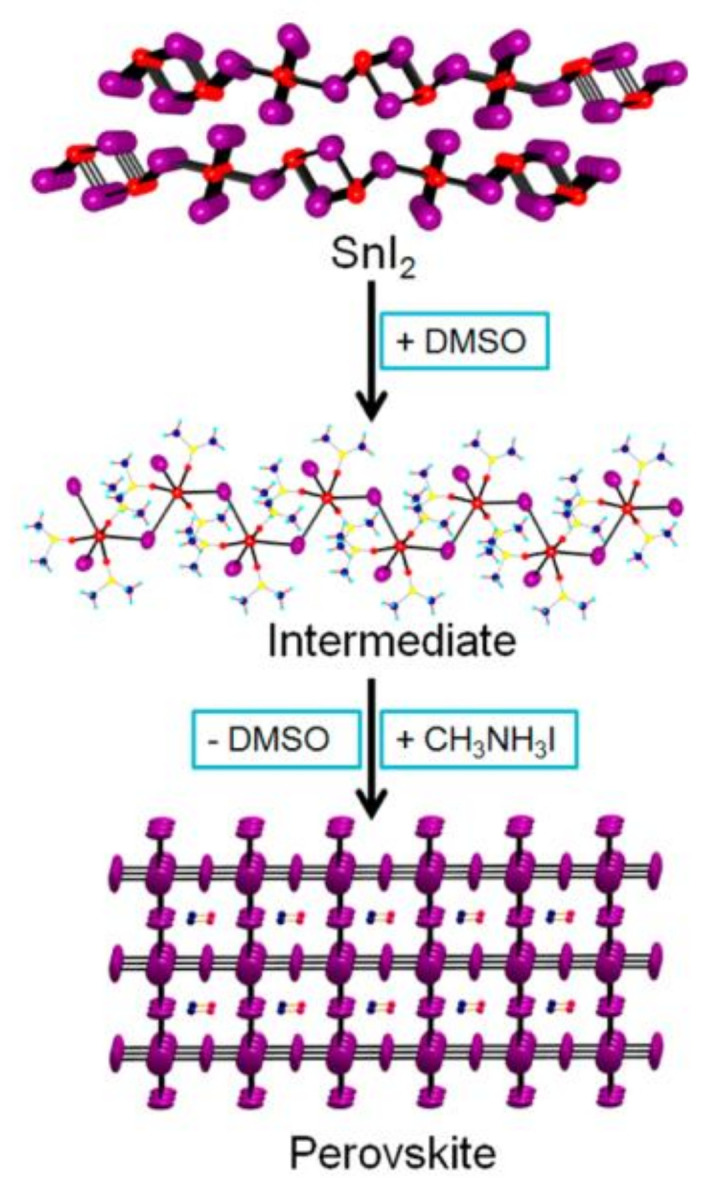
A schematic on the film formation of the CH_3_NH_3_SnI_3_ perovskite film starting from SnI_2_ through the SnI_2_·3DMSO intermediate. Reproduced with permission from [32]. Copyright American Chemical Society, 2015.

**Figure 4 nanomaterials-13-00585-f004:**
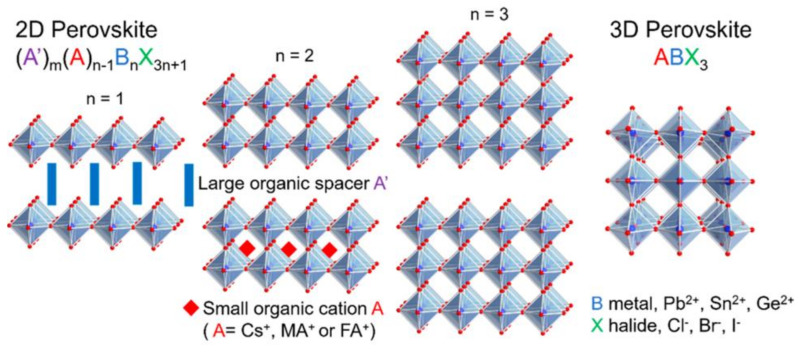
Schematic illustration of the evolution from 2D perovskite to 3D perovskite with key components. Reproduced with permission from [63]. Copyright American Chemical Society, 2019.

**Figure 5 nanomaterials-13-00585-f005:**
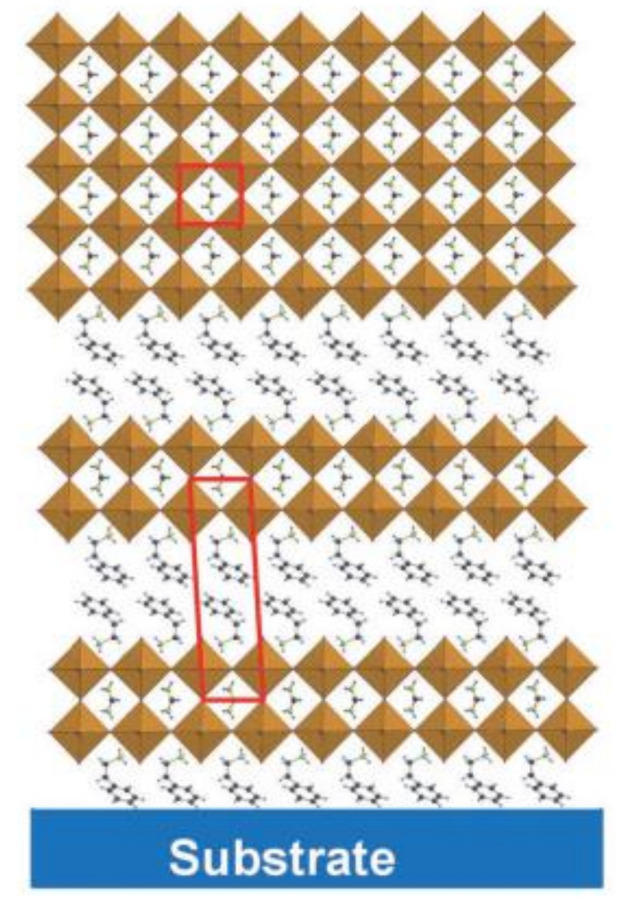
Schematic illustration of 2D/3D mixture. Reproduced with permission from [64]. Copyright John Wiley and Sons, 2017.

**Figure 6 nanomaterials-13-00585-f006:**
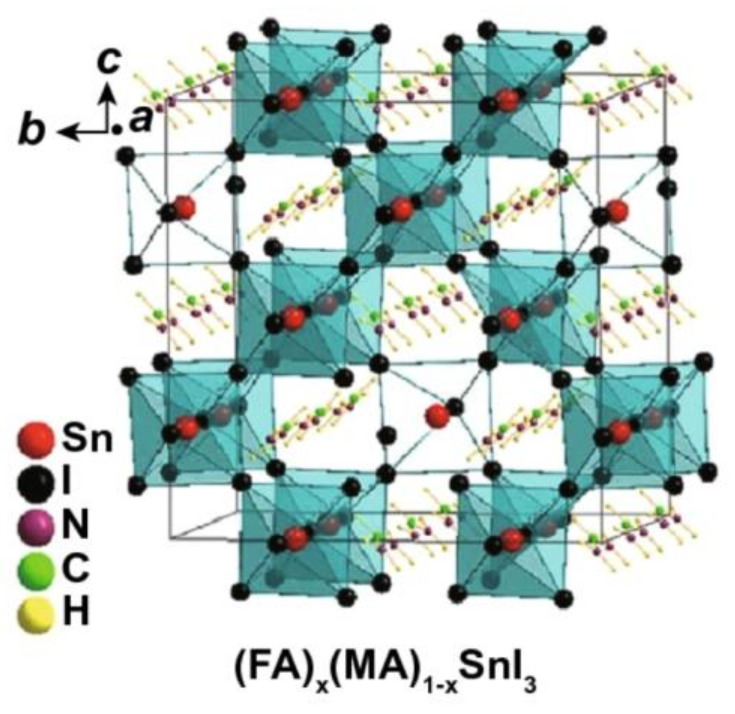
Schematic illustration of (FA)_x_(MA)_01-x_SnI_3_. Reproduced with permission from [38]. Copyright Springer Nature, 2021.

**Table 1 nanomaterials-13-00585-t001:** Characteristics of MASnI_3_ and MAPbI_3_.

Characteristics	MASnI_3_	MAPbI_3_
Charge mobility (cm^2^ V^−1^ s^−1^)	10^2^–10^3^	10–10^2^
Direct bandgap (eV)	1.1	1.5
Absorption coefficient (cm^−1^)	1.82 × 10^4^	1.80 × 10^4^

**Table 2 nanomaterials-13-00585-t002:** Performance of MASnX_3_.

Perovskite	Device Architecture	V_OC_ (V)	Jsc (mA cm^−2^)	FF (%)	PCE (%)	Year	References
MASnI_3_	c-TiO_2_/m-TiO_2_/Perovskite/spiro-MeOTAD/Au	0.68	16.3	48	5.23	2014	[24]
MASnI_2_Br	c-TiO_2_/m-TiO_2_/Perovskite/spiro-MeOTAD/Au	0.77	14.38	50	5.48	2014	[24]
MASnIBr_2_	c-TiO_2_/m-TiO_2_/Perovskite/spiro-MeOTAD/Au	0.82	12.3	57	5.73	2014	[24]
MASnBr_3_	c-TiO_2_/m-TiO_2_/Perovskite/spiro-MeOTAD/Au	0.88	8.26	59	4.27	2014	[24]
MASnI_3_	TiO_2_/Perovskite/spiro-OMeTAD/Au	0.716	15.18	50.07	5.44	2014	[36]
MASnI_3_	FTO/compact TiO_2_/mp-TiO_2_/Perovskite /PTAA/Au	0.342	19.97	63	6.83	2019	[37]

**Table 3 nanomaterials-13-00585-t003:** Performance of FASnX_3_.

Perovskite	Device Architecture	V_OC_ (V)	Jsc (mA cm^−2^)	FF (%)	PCE (%)	Year	References
FASnI_3_ + 20 mol% SnF_2_	TiO_2_/Perovskite/spiro-OMeTAD/Au	0.238	24.45	36	2.1	2015	[42]
FASnI_2_Br	ITO/PEDOT:PSS/Perovskite/C_60_/Ca/Al	0.467	6.82	54.3	1.72	2016	[49]
FASnI_3_ + 10 mol% SnF_2_	ITO/PEDOT:PSS/ Perovskite/C_60_/BCP/Ag	0.46	20.68	58.88	5.59	2016	[39]
FASnI_3_ + 15 mol% SnF_2_	ITO/PEDOT:PSS/ Perovskite/C_60_/BCP/Ag	0.43	18.74	57.21	4.61	2016	[39]
FASnI_3_ + 20 mol% SnF_2_	ITO/PEDOT:PSS/ Perovskite/C_60_/BCP/Ag	0.393	19.12	54.12	4.06	2016	[39]
FASnI_3_ + 30 mol% SnF_2_	ITO/PEDOT:PSS/ Perovskite/C_60_/BCP/Ag	0.34	18.63	52.83	3.35	2016	[39]
FASnI_3_ + SnF_2_-pyrazine	FTO/bl-TiO_2_/mp-TiO_2_/Perovskite/Spiro-OMeTAD/Au	0.2833	23.44	55.56	3.708	2016	[43]
FASnI_3_ + 10% en	FTO/c-TiO_2_/m-TiO_2_/{en}FASnI_3_ /PTAA/Au	0.48	22.54	65.96	7.14	2017	[29]
FASnI_3_	ITP-coated Glass/PEDOT:PSS/Perovskite/doped C_60_	0.29	5.33	33.8	0.52	2017	[40]
FASnI_3_ + SnF_2_	ITP-coated Glass/PEDOT:PSS/Perovskite/doped C_60_	0.4	17.89	58.8	4.2	2017	[40]
FASnI_3_ + SnF_2_ + TMA	ITP-coated Glass/PEDOT:PSS/Perovskite/doped C_60_	0.47	22.45	67.8	7.09	2017	[40]
FASnI_3_ + 5 mol% AHP	ITO/CuSCN/Perovskite/PCBM/Ag	0.55	19.39	68.82	7.34	2019	[17]
FASnI_3_: π-conjugated Lewis base	ITO/PEDOT:PSS/ Perovskite/C_60_/BCP/Ag	0.63	21.22	74.7	10.1	2019	[47]
FASnI_3_ + 50% LFA	ITO/PEDOT:PSS/Perovskite/C_60_/BCP/Ag	0.628	22.25	74.2	10.37	2020	[50]

**Table 4 nanomaterials-13-00585-t004:** Performance of CsSnX_3_.

Perovskite	Device Architecture	V_OC_ (V)	Jsc (mA cm^−2^)	FF (%)	PCE (%)	Year	References
CsSnI_3_ + 20 mol % SnF_2_	FTO/c-TiO_2_/ m-TiO_2_/Perovskite/HTM/Au	0.24	22.70	37	2.02	2014	[53]
CsSnI_3_ + 10% SnI_2_	ITO/CuI/Perovskite/C_60_/BCP/Al	0.55	8.5	55	2.1	2015	[11]
CsSnI_3_	c-TiO_2_/Perovskite/spiro-MeOTAD/Au	0.86	23.2	65	12.96	2016	[62]
CsSnBr_3_	c-TiO_2_/Perovskite/spiro-MeOTAD/Au	0.85	21.23	58	10.46	2016	[62]
CsSnCl_3_	c-TiO_2_/Perovskite/spiro-MeOTAD/Au	0.87	19.82	56	9.66	2016	[62]
Cs_2_SnI_6_	FTO/TiO_2_/Perovskite /P3HT/Ag	0.51	5.41	35	0.96	2017	[13]
CsSnI_3_ + PTM	ITO/PEDOT:PSS/Perovskite/PCBM/BCP/Ag	0.64	21.81	72.1	10.1	2021	[61]

**Table 5 nanomaterials-13-00585-t005:** Performance of quasi-2D and 2D/3D mixed perovskites.

Perovskite	Device Architecture	V_OC_ (V)	Jsc (mA cm^−2^)	FF (%)	PCE (%)	Year	References
(BA_0.5_PEA_0.5_)_2_FA_3_Sn_4_I_13_	ITO/PEDOT:PSS/Perovskite/TFB/MoO3/Au	0.6	21.82	66.73	8.82	2019	[73]
PEA_x_FA_1_−xSnI_3_ + NH_4_SCN	PEDOT:PSS/Perovskite/ICBA/BCP/Ag	0.94	17.4	75	12.4	2020	[75]

**Table 6 nanomaterials-13-00585-t006:** Performance of mixed A cations Sn-based perovskites.

Perovskite	Device Architecture	V_OC_ (V)	Jsc (mA cm^−2^)	FF (%)	PCE (%)	Year	References
(FA)_0.75_(MA)_0.25_SnI_3_ + 10% SnF_2_	PEDOT:PSS/Perovskite/C_60_/BCP/Ag	0.61	21.2	62.7	8.12	2017	[59]
(FA)_0.5_(MA)_0.5_SnI_3_ + 10% SnF_2_	PEDOT:PSS/Perovskite/C_60_/BCP/Ag	0.53	21.3	52.4	5.92	2017	[59]
(FA)_0.25_(MA)_0.75_SnI_3_ + 10% SnF_2_	PEDOT:PSS/Perovskite/C_60_/BCP/Ag	0.48	20.7	45.2	4.49	2017	[59]

## Data Availability

Not applicable.

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
