# Peer review of "A Review on the Progress, Challenges, and Performances of Tin-Based Perovskite Solar Cells"

_nanomaterials, 2023, doi:10.3390/nano13030585_

Round 1
Author Response
Dear Reviewer,
Thank you for your time and patience in going through our manuscript. We have revised the manuscript, and here is the response to the comment:
- The literature suggested has been cited in the revised manuscript.
- The PCE value of the Sn-based PSCs has been updated with recent references.
- The introduction section has been modified according to the reviewer’s comments.
Your sincerely,
Lye Yuen Ean
Reviewer 2 Report
The manuscript entitled “Lead-free perovskite solar cells: A review on the progress, challenges, and performances of tin-based perovskite solar cells” by Ng et al summarized numerous types of Sn-based perovskite with important achievements on experimental-based studies to date. However, as a review, there are still a number of obvious issues that need to be addressed.
1. The manuscript lacked sufficient figures to show the structure and properties of various types of Sn-based perovskite.
2. The text in the Figure 1.1 was too small and difficult to read.
3. The manuscript lacked discussions of Sn-based perovskite preparation.
4. Various Pb-free PSCs have made significant progress, including tin (Sn), bismuth (Bi), germanium (Ge), and antimony (Sb). There needs to be tables and discussions to show their advantages and disadvantages.
5. Some relevant literature needs to be cited, such as Energy Environ. Sci., 2020, 13, 2363-2385; Joule, 2021, 5, 863-886; Adv. Energy. Mater., 2022, 12, 202202491.
Author Response
Dear Reviewer,
Thank you for your time and patience in going through our manuscript. We have revised the manuscript, and here is the response to the comment:
- The properties of the Sn-based perovskite have been tabulated in Tables 2.2, 3.1,4.1, 5.1, and 6.1. Figure 1 was modified by labelling the Sn material in the active layer.
- Figure 1.1 has been removed from the manuscript.
- The manuscript focusses on the recent progress, challenges, and performance of Sn-based perovskite. So, the preparation process was not included in the manuscript.
- Yes, other alternatives could replace the lead in PSC, but tin (Sn) has exhibited a significant PCE among these alternatives. Thus, a detailed review of the performance of Sn-based perovskite was conducted.
- Similar literature has been cited in the following section. For example, Wang, M.; Wang, W.; Ma, B.; Shen, W.; Liu, L.; Cao, K.; Chen, S.; Huang, W. Lead-Free Perovskite Materials for Solar Cells. Nano-Micro Letters2021, 13, doi:10.1007/s40820-020-00578-z.
Your sincerely,
Lye Yuen Ean

Reviewer 3 Report
The authors present an overview of Sn-based halide perovskite assessing the current challenges in photovoltaic applications and possible remedies. An assessment of characteristics, progress, and performance of different types of Sn-based halide perovskites are also presented. While I think the area is very important and exciting for future development of lead-free perovskite solar cells, the manuscript fails to deliver the importance of this specific review. The review feels quite generic in nature without any clear objective as well as definite structure. There are already several review discussing the fate and future of Sn-based halide perovskites (such as Adv. Energy Mater. 2022, 2202209; Small Struct., 3: 2100102; Energy Environ. Sci., 2021, 14, 4292-4317 ). The authors should focus on the specific question for which their viewpoint is important and deserves publication. Specific comments and suggestions are given below:
1. The title of the manuscript is misleading as it is primarily focused on Sn-based halide perovskites. On the other hand, lead-free halide perovskites encompasses a much wider field. I would recommend to omit the first part of the title. Authors could focus on Sn-based perovskites in the abstract as the importance is somewhat diluted at current form.
2. The authors need help in improving the language throughout the manuscript as in many sections the text and phrases are unclear or confusing. For example, ABX3 is not the formula of perovskite solar cells, rather it’s the formula of the perovskite compounds. When perovskite compound is used as photoactive layer, it’s called perovskite solar cell. “A” is not only organic cation, but can be organic, inorganic, or a combination of both. Later, the authors used the term “ideal optical characteristics”, what does it signify? Ideal for which application? “light-absorbing materials in LEDs” should be light-emitting materials. There are many more confusing texts and phrases throughout the manuscript.
3. The overarching message of the review articles should be made much clearer. The structure of the manuscript could be more simpler which basic introduction of the lead-free halide perovskite in general could be merged with “Introduction” section. The current challenges can be separated based on key characteristics, such as presence of defects, deposition techniques, or stability etc.
4. I think the readers would greatly appreciate the explanation of the key terminologies used in this manuscript, like what is conventional or pin structure? What are the important characteristics of photoactive materials and how they are compared to existing semiconductors.
5. The figure 1 fails to provide enough context and accurate information regarding the pin or nip configuration. The authors should clarify which element sits on top or bottom of perovskite layer for which configurations. As a reader, I am sure how to read that figure. I would also suggest the authors to put few figures to illustrate the key concept of the manuscript especially in the challenges section. This would greatly help the reader to inter-connect to each other and visualize the bigger picture.
6. On a smaller note, the number of many references are missing in the text. All the tables should be in chronological order. The oxidation state of an elements is usually written as superscript.
Author Response
Dear Reviewer,
Thank you for your time and patience in going through our manuscript. We have revised the manuscript, and here is the response to the comment:
- The title of the manuscript has been revised to be more specific.
- Changes have been made in explaining the formula of metal halides perovskite. The light-absorbing material refers to the perovskite not LED in the content.
- Changes have been made according to the comment.
- The characteristics of 3G solar cells have been mentioned in the introduction section. An explanation of layers in the cell architecture was included.
- The figure has been split into two for better understanding.
- Some of the references were not mentioned in the text because their fabrication method was similar. Their performance was collected for comparison purposes. Changes have been made according to the comment.
Your sincerely,
Lye Yuen Ean
Round 2
Reviewer 2 Report
Thanks to the authors for their responses and modifications.
I insist that the manuscript lacked sufficient figures to show the structure and properties of various types of Sn-based perovskite.
The authors should add no less than 5 relevant figures to illustrate the review content to make it easier for readers to understand, so that the manuscript meet the requirements for publication!
Author Response
Dear Reviewer,
Thank you for your valuable comment. We have revised the manuscript, and relevant figures were added to the manuscript. The chemical structure for MaSnX3, FaSnX3, and CsSnX3 were the same, so a general perovskite structure was added to the introduction.
Reviewer 3 Report
The authors have well addressed my questions. Nevertheless, I still have couple of suggestions:
1. Some of the citations are still missing (page 3. line 169-170; page 5, line 237-238; page 6, line 276, page 8, line 350, and few more)
2. Authors should also include recent literature on CsSnI3-based solar cells. There are several reports after 2016 which also need to be mentioned.
Author Response
Dear Reviewer,
Thank you for your valuable comment. We have revised the manuscript, the missing citations have been added, and recent literature on CsSnI3-based solar cells was included in the manuscript.